# GRKs as Modulators of Neurotransmitter Receptors

**DOI:** 10.3390/cells10010052

**Published:** 2020-12-31

**Authors:** Eugenia V. Gurevich, Vsevolod V. Gurevich

**Affiliations:** Department of Pharmacology, Vanderbilt University, Nashville, TN 27232, USA; vsevolod.gurevich@vanderbilt.edu

**Keywords:** GRK, GPCR, neurotransmitter, arrestin

## Abstract

Many receptors for neurotransmitters, such as dopamine, norepinephrine, acetylcholine, and neuropeptides, belong to the superfamily of G protein-coupled receptors (GPCRs). A general model posits that GPCRs undergo two-step homologous desensitization: the active receptor is phosphorylated by kinases of the G protein-coupled receptor kinase (GRK) family, whereupon arrestin proteins specifically bind active phosphorylated receptors, shutting down G protein-mediated signaling, facilitating receptor internalization, and initiating distinct signaling pathways via arrestin-based scaffolding. Here, we review the mechanisms of GRK-dependent regulation of neurotransmitter receptors, focusing on the diverse modes of GRK-mediated phosphorylation of receptor subtypes. The immediate signaling consequences of GRK-mediated receptor phosphorylation, such as arrestin recruitment, desensitization, and internalization/resensitization, are equally diverse, depending not only on the receptor subtype but also on phosphorylation by GRKs of select receptor residues. We discuss the signaling outcome as well as the biological and behavioral consequences of the GRK-dependent phosphorylation of neurotransmitter receptors where known.

## 1. Introduction: GRKs in GPCR Signaling and Trafficking

Activation-dependent phosphorylation of rhodopsin was discovered in the early 1970s [1,2], long before it became clear that rhodopsin belongs to the family of G protein-coupled receptors (GPCRs). Comparison of the primary structure and membrane organization of rhodopsin [3] and β2-adrenergic receptor (β2AR) [4] demonstrated that both belong to the same protein family, now known as the GPCR superfamily, of which humans express ~800 different subtypes. These receptors are also called 7TMRs, as all GPCRs contain seven transmembrane α-helices. Rhodopsin kinase (modern systematic name GRK1, which stands for G protein-coupled receptor kinase 1 [5]) was the first GPCR kinase (GRK) discovered. It was shown to selectively bind and phosphorylate light-activated rhodopsin [6,7]. Later, the Kuhn group demonstrated that rhodopsin phosphorylation is necessary to quench its signaling [8]. In the same year, the Lefkowitz group reported that β2AR is phosphorylated by a cAMP-independent kinase and that this phosphorylation facilitates receptor desensitization [9]. As this kinase was discovered via its ability to phosphorylate β2AR, when cloned, it was called β2-adrenergic receptor kinase, or βARK for short (modern systematic name GRK2) [10]. GRK2 was also shown to phosphorylate rhodopsin in a strictly activation-dependent manner, exactly like rhodopsin kinase [11], suggesting that both GRKs are specific for active GPCRs. When rhodopsin kinase was cloned [12], it was found to have a structure similar to that of the kinase that phosphorylates agonist-activated β2AR [10]. The mechanism underlying the specificity of GRKs for active GPCRs was subsequently established: the activation of GRK1 required physical interaction with the active rhodopsin [13]. The same activation mechanism was demonstrated for non-visual GRKs with β2AR [14] and D2R [15].

Rhodopsin phosphorylation was shown to increase the binding of a 48-kDa protein [16] (later termed visual or rod arrestin; systematic name arrestin-1). The binding of this protein was shown to be necessary to stop rhodopsin coupling to its cognate G protein, transducin [8]. It turned out that highly purified β2AR kinase has a limited effect on β2AR coupling to its cognate G protein, Gs, suggesting that a non-visual arrestin homologue is necessary for desensitization [17]. Both arrestin-1 [18] and its non-visual homologue [19] were subsequently cloned. These proteins are highly homologous. However, arrestin-1 and its homologue demonstrated clear preference for rhodopsin and β2AR, respectively [20]. Therefore, the non-visual protein was originally termed β-arrestin (systematic name arrestin-2). 

In genetically modified mice, it was shown that the absence of either GRK1 [21] or arrestin-1 [22] results in abnormally prolonged rhodopsin signaling. These and other findings resulted in the general model of two-step homologous (specific for the receptor that was activated) GPCR desensitization: the active receptor is phosphorylated by a GRK (seven GRK subtypes are expressed in most vertebrates [5,23]), whereupon arrestin (four subtypes are expressed in most vertebrates [24]) binds to it and stops the signaling by direct competition with the G protein [25,26] (reviewed in [27]) (Figure 1). Non-visual arrestins bound to phosphorylated GPCRs were shown to directly interact with the key components of the internalization machinery of the coated pit, clathrin [28], and clathrin adaptor AP2 [29], so that GPCR phosphorylation and subsequent arrestin binding promote receptor endocytosis. While the role of both GPCR phosphorylation and arrestin binding actually differs for various receptor subtypes (Figure 2), the simplified general paradigm posits that GRKs play a critical role in the two processes that reduce cell responsiveness: precluding G protein coupling and facilitating receptor removal from the plasma membrane. These are the key GRK effects on the signaling of most GPCRs, including the neurotransmitter receptors belonging to the GPCR family that are discussed below. The amount of information available regarding the effects of GRK phosphorylation, particularly in biologically relevant in vivo models, varies significantly for different receptors. While several independent studies have been performed with opioid receptors, including the use of knockin mice expressing receptor mutants where some or all potential GRK phosphorylation sites were eliminated, other neurotransmitter GPCRs have been studied much less comprehensively. As this review is based on experimental evidence, opioid receptors will be discussed in greater detail than other GPCRs that bind neurotransmitters. 

## 2. Regulation of Non-Visual GPCRs by GRK Phosphorylation

Even the briefest survey of the mechanisms of GRK-dependent regulation of GPCR signaling in general and of neurotransmitter receptors in particular gives an impression of an almost unlimited variety. This is not surprising, since GPCR structural homology, even within the transmembrane domains, is quite limited. Homology is virtually nonexistent in the intracellular elements such as the 3rd loop or C-terminus, which in most receptors harbor the GRK phosphorylation sites [35] (Figure 2). The number and position of potential GRK targets differs significantly even among members of the same GPCR subfamily, let alone between subfamilies. Therefore, the behavior of every GPCR with regard to GRK-dependent phosphorylation and its effects on G protein coupling, subsequent arrestin binding, and arrestin-mediated signaling is different and has to be studied as such, instead of relying on information obtained with “model” receptors. This is particularly important for clinically relevant GPCRs, which are targeted by numerous therapeutically important drugs. 

### 2.1. The Model GPCRs: Adrenergic Receptors

Adrenergic receptors have long served as prototypical non-visual GPCRs in studies of signaling regulation. Adrenergic receptors bind catecholamines, epinephrine (adrenaline), norepinephrine (noradrenaline), and dopamine (with lower affinity). The three β-adrenergic receptors, β1, β2 (β2AR), and β3, couple to Gs, whereas alpha-adrenergic α1 receptors predominantly couple to Gq/11 and α2 receptors preferentially couple to members of the Gi/o subfamily of G proteins. The role of GRKs in the regulation of non-visual GPCRs was first discovered using β2AR as a model receptor (Figure 2). It was shown that cAMP-activated protein kinase A (PKA) phosphorylates β2AR on the third cytoplasmic loop, whereas the eight GRK phosphorylation sites are localized in the receptor C-terminus. Rapid desensitization of β2AR requires the action of both kinases [36]. Interestingly, similar phosphorylation by both classes of kinases mediates the desensitization of the closely related β1AR [37], but the mechanisms whereby the two kinases exert their effects are somewhat different. PKA phosphorylation switches β2AR, but not β1AR, preference from the cyclase activating Gs to the inhibitory Gi [38,39], whereas GRK phosphorylation in both cases promotes the binding of arrestins [17], which block G protein coupling by direct competition [25,26]. The role of PKA phosphorylation is controversial, as there are reports of PKA phosphorylation switching β1AR coupling from Gs to Gi [40], as well as studies suggesting that upon PKA phosphorylation β2AR does not switch to Gi [41,42]. While both β1AR and β2AR have GRK phosphorylation sites in their C-termini, β3AR has a very short C-terminus lacking GRK targets [43]. Indeed, β3AR was not phosphorylated upon agonist treatment and did not undergo the rapid desensitization and internalization characteristic for β2AR [43]. Importantly, when β3AR was equipped with the β2AR C-terminus, the chimera was regulated essentially like β2AR [43]. Thus, β3AR is not subject to rapid GRK/arrestin-mediated desensitization. This makes sense biologically, as β3AR is mostly expressed in brown fat cells, and the regulation of their metabolism is tonic rather than acute.

Both subfamilies of α-adrenergic receptors are regulated by GRKs and arrestins, but α1 and α2 are structurally different. While α1 receptors resemble β-adrenergic receptors, with a relatively short third cytoplasmic loop and a C-tail that contains GRK phosphorylation sites, α2 receptors have very long third cytoplasmic loops that contain serine (Ser) and threonine (Thr) residues targeted by GRKs [44]. Interestingly, α2 receptors were the first where it was shown that GPCR phosphorylation depends on the ability of the receptor to bind and activate GRKs, rather than on the presence of GRK phosphorylation sites on their cytoplasmic elements. The α2A receptor is phosphorylated by GRKs2/3 in the third loop and undergoes rapid desensitization [45,46,47]. In contrast, the α2C receptor is resistant to desensitization in spite of the presence of multiple serines and threonines as potential phosphorylation sites in the third cytosolic loop [44,46,47]. Experiments with chimeric receptors have demonstrated that when the third loop of the α2A receptor was replaced with the third loop of the α2C receptor, the chimera was rapidly phosphorylated and desensitized, whereas the reverse chimera, the α2C receptor with the third loop from α2A, was not. These experiments clearly demonstrated that the problem was not the loop itself, which was perfectly capable of being phosphorylated by GRKs, but rather a specific receptor conformation induced by agonists in the α2C receptor that is not conducive to GRK binding and/or activation [44].

### 2.2. Dopamine Receptors: Outside of the Classical Paradigm? 

Mammals express five dopamine receptors, D1, D2, D3, D4, and D5. Both D1 and D5 couple to the adenylyl cyclase-activating Gs, whereas D2, D3, and D4 couple to the Gi/o subfamily of cyclase-inhibiting G proteins. Structurally, both Gs-coupled D1 and D5 have a relatively short third cytoplasmic loop and a fairly long C-terminus with numerous potential GRK phosphorylation sites, whereas Gi/o-coupled D2, D3, and D4 have a much larger third cytoplasmic loop with potential phosphorylation sites and a very short C-terminus [48]. D1 and D2 receptors robustly bind both non-visual arrestins [49]. The regulation of dopamine receptors by GRK isoforms has previously been reviewed in detail [50].

The rat D1 receptor has eight potential GRK phosphorylation sites in the third loop and 18 in the C-terminus (Figure 2). Mutational analysis of these sites suggested that their phosphorylation might proceed in a hierarchical manner, with the C-terminus being phosphorylated first, followed by the third loop [33]. It seems that phosphorylation in the C-terminus or third loop is not required for arrestin binding per se but rather serves to promote the dissociation of these elements and thus allows arrestin to bind by eliminating steric hindrance [33]. This conclusion is based on the fact that D1 receptor mutants with a truncated C-terminus undergo normal desensitization and recruit arrestins even when phosphorylation of the third loop residues is completely abrogated. At the same time, when the third loop phosphorylatable residues are mutated in a receptor with an intact C-terminus, arrestin recruitment is impaired, indicating that third loop phosphorylation, while not directly needed for arrestin binding, does serve a purpose. Although in vitro, the D1 receptor is readily phosphorylated by all four ubiquitous GRK isoforms [51,52], it remains unknown which GRK is involved in each phosphorylation event. A recent spree of development of biased ligands for D1 receptors yielded several classes of orthosteric compounds that activated D1 receptor-mediated Gs signaling but did not induce arrestin recruitment and receptor desensitization [53,54]. Interestingly, some of these drugs display bias only for the D1 receptor, while acting as unbiased agonists at the closely related D5 receptor [53]. Unfortunately, receptor phosphorylation was not examined in these studies. It would be interesting to see whether these agonists fail to promote the receptor conformation favorable for GRK binding/activation (as in α2C receptor described above) and, consequently, receptor phosphorylation and arrestin recruitment, or whether they act via some other mechanism.

The second main dopamine receptor subtype, the D2 receptor, has a very short C-terminus but an extra-long third loop (Figure 2). It has three threonines in the first loop, two serines and two threonines in the second, and 13 serines and 10 threonines in the third loop. Upon agonist stimulation, the D2 receptor undergoes GRK-mediated phosphorylation and recruits arrestins [55,56]. Elimination of all putative phosphorylatable residues (mutations to alanines) in the second and third loops eliminated GRK-dependent phosphorylation and reduced, but did not abolish, receptor internalization, while arrestin recruitment seemed to be preserved [56]. Another study has mapped six serine and six threonine sites in the third loop as the sites for GRK2/3-dependent phosphorylation and confirmed that phosphorylation is not required for arrestin binding, receptor internalization, or desensitization [34]. Curiously, the D2 receptor is very slow to desensitize, requiring hours of agonist treatment [56,57], whereas GRK/arrestin-dependent desensitization following the general paradigm (Figure 1) usually occurs within minutes. GRK2 may promote D2 receptor desensitization in a phosphorylation-independent manner: its effect was preserved in the mutants lacking the second/third loop phosphorylation sites, and the kinase-dead GRK2-K220R mutant had the same effect as GRK2 [56]. Others found that phosphorylation by GRKs was required for desensitization, but not the phosphorylation of the receptor itself [58]. Stimulation by dopamine induces desensitization of the D2 receptor-dependent GIRK current in Xenopus oocytes in a GRK-independent, arrestin-dependent manner [59]. However, GRK-dependent phosphorylation was shown to be required for receptor resensitization and recycling, which was significantly impaired for the receptor lacking GRK sites [34,56]. Even though GRK2 often binds the Gβγ released upon G protein activation and uses it as a membrane anchor [60,61,62], D2R can recruit GRK2 directly without G protein activation, as revealed by studies using the arrestin-preferring D2R mutant and the arrestin-biased agonist UNC9994 [15]. It appears safe to conclude that phosphorylation of the D2 receptor by GRKs is not needed for its desensitization and, probably, internalization, but might be required for the receptor to be recycled and resensitized. Studies with GRK knockout mice showed that only mice lacking GRK6 demonstrate behavioral supersensitivity to the indirect dopaminergic drugs amphetamine and cocaine, presumably due to defective desensitization of dopamine receptors [63]. This effect was attributed to the action of GRK6 at the D2 receptor. This is contrary to the data obtained in cultured cells showing that only GRKs 2 and 3 phosphorylate the D2 receptor [34,52]. Additionally, it is unclear how to reconcile the in vivo data with the evidence that the D2 receptor is resistant to desensitization, and when it desensitizes, it does so independently of phosphorylation by GRKs [55,56,57,58,59].

The D3 dopamine receptor belongs to the same subfamily as D2 but displays unique signaling properties. It has a significantly higher affinity for dopamine than D2, a minimal agonist-induced shift to the high-affinity state, and a weak ability to activate Gi/o-dependent signaling [64,65]. When it was first cloned, the D3 receptor attracted significant attention as a potential target for antipsychotic drugs due to its selective expression in the limbic area of the brain [64,65,66,67]. The D3 receptor is resistant to GRK-mediated phosphorylation and shows minimal arrestin recruitment and internalization, with all these processes facilitated upon GRK2 overexpression [55]. It has been suggested that desensitization of the D3 receptor occurs in a unique manner termed pharmacological sequestration. This mechanism involves translocation of the receptor to more hydrophobic domains in the plasma membrane without removal into the intracellular compartments, and it does require arrestins [68]. The D3 autoreceptor exists in a complex with filamin-A, which enhances D3 coupling to G protein. Agonist stimulation reduced the D3 receptor association with filamin-A, thereby desensitizing the receptor, and so did overexpression of GRK2/3, which facilitated agonist-induced recruitment of arrestin [69]. Although suggesting a role for GRK2/3 in D3 receptor desensitization, this study has not determined the phosphorylation site(s) and did not show whether GRK-dependent receptor phosphorylation occurred upon agonist exposure. A class of recently developed D3 receptor G protein-biased agonists do not cause D3 receptor desensitization but do induce its internalization (as opposed to pharmacological sequestration caused by dopamine) in a GRK2-dependent but arrestin-independent manner [70,71]. It remains unknown whether GRK2 phosphorylates the D3 receptor in this case and whether or how this phosphorylation changes the receptor behavior.

### 2.3. Muscarinic Acetylcholine Receptors: Phosphorylation Sites Come in Clusters 

All five members of the muscarinic receptor subfamily are GPCRs [72] subject to GRK phosphorylation. While M1, M3, and M5 predominantly couple to Gq/11, both M2 and M4 couple to the Gi/o subfamily of G proteins [73]. All muscarinic receptors have a large third cytoplasmic loop (157–240 residues), similar to α2 receptors. This loop contains GRK phosphorylation sites, and GRK phosphorylation plays an important role in the regulation of muscarinic receptor signaling [74].

Phosphorylation and arrestin binding have been extensively studied for the M2 muscarinic receptor. The M2 receptor has a very long third cytosolic loop that possesses multiple Ser/Thr sites (44 in the human M2) and a very short C-terminus with no potential GRK phosphorylation sites (Figure 2). Earlier studies have demonstrated that the M2 receptor is phosphorylated upon agonist activation in the central part of the third loop (residues 250–323), which harbors 25 of the phosphorylatable serines and threonines [31,60]. Alanine substitutions of eight residues in the clusters Ser286-Ser290 and Thr307-Ser311 located in that part of the third loop abolished agonist-dependent phosphorylation, which, however, was preserved if only one of the clusters was mutated. The receptor desensitization was abolished by substitution of the Thr307-Ser311 cluster, termed “C-terminal”, but not of the “N-terminal” cluster or of other Ser/Thr residues [75,76]. Mutation of the C-terminal cluster alone did not abolish the receptor internalization, whereas mutation of both clusters resulted in severely impaired desensitization and internalization. Thus, the N-terminal cluster acts as a “brake”, so that its elimination does not affect arrestin binding much, whereas a phosphorylated C-terminal cluster is required for arrestin binding. The M2 receptor is phosphorylated well by GRK2 and GRK3, which are biochemically quite similar [30,31,52]. The question of whether the M2 receptor is a substrate for other GRK isoforms remains open. It has been shown to be phosphorylated by both GRK5 and 6 in cultured cells but with low efficacy [52]. Mice lacking GRK5 show behavioral supersensitivity to the nonselective muscarinic agonist oxotremorine [77]. Since M2 receptors mediate many of the central behavioral effects of cholinergic drugs [78,79], this could be indicative of the role of GRK5 in the desensitization of the M2 receptor in vivo. Furthermore, GRK5 knockout mice demonstrated resistance to smooth muscle relaxation induced by the β-adrenoreceptor agonist isoproterenol, the effect mediated by M2 receptors in airway smooth muscle [80]. This is another indication of a role for GRK5 in the regulation of M2 receptor signaling in vivo. Mice lacking GRK6 show no supersensitivity to cholinergic stimulation [63,81], suggesting that GRK6 is unlikely to be involved in the regulation of the M2 receptor in the brain.

Curiously, while GRKs play a clear role in the regulation of M2 signaling [74], GRK-mediated phosphorylation was shown to increase arrestin binding to M2 to a lesser extent than to β2AR [82]. This suggests a functional role of GRK phosphorylation of receptor or non-receptor substrates independent of promoting arrestin binding. Even though arrestins robustly bind phosphorylated M2 receptor [83,84,85] and this is important for the termination of its G protein-mediated signaling, the M2 receptor internalizes via an arrestin- and dynamin-independent mechanism [75], which still remains to be elucidated. Later work highlighted the importance of the third loop in directing the M2 receptor towards dynamin-independent internalization [86,87] and identified a specific sequence in the loop responsible for this effect [86]. Interestingly, the M2 receptor does not recycle following internalization [86,87], whereas the related M4 receptor, which is internalized via the dynamin-dependent clathrin pathway [87,88], as well as the chimeric M2 with the third loop from M4, do recycle [86,87,88]. Curiously, observations in mouse hippocampal neurons demonstrated agonist-dependent internalization of the M2 receptor via clathrin coated pits, contrary to what has been observed in cultured cells [89]. 

A member of another subfamily of muscarinic receptors, the M1 receptor, which is a major postsynaptic muscarinic receptor throughout the brain, is phosphorylated in an agonist-dependent manner by GRK2 at sites located in the middle of the third loop [90]. Different deletions in the central portion of the third loop resulted in impairment of the M1 internalization ranging from complete to partial (10–30%) [91]. Unfortunately, specific phosphorylation sites and the mode of phosphorylation of the M1 receptor have never been determined. The deletion of a single element ^284^Ser-Glu^292^ in the third loop leads to a defect in internalization [91]. However, judging by the magnitude of this defect, this element is obviously only one of several elements involved in receptor internalization. The specific effects of these phosphorylation events on arrestin binding and the role of arrestin binding in M1 receptor desensitization/internalization remain largely unexplored. It has been shown that upon agonist stimulation, arrestin is recruited to the M1 receptor, promoting its internalization. The third loop plays the primary role in this process, as deletion of the loop severely impairs both arrestin recruitment and internalization [92]. The study did not directly investigate the relationship between the GRK-dependent phosphorylation of the third loop sites and arrestin recruitment. The use of GRK2 inhibitor showed that recruitment of arrestin-3 to the M1 receptor is partially dependent on GRK2-mediated phosphorylation [92]. The M1 receptor in the hippocampal neurons appears to be primarily phosphorylated by GRK2, which induces receptor desensitization, but sequestering of active Gαq via the GRK2 RGS homology domain (reviewed in [5]) contributes to the receptor desensitization in a phosphorylation-independent manner [93,94]. This study highlighted the role of the RGS homology domain of GRKs 2/3 in the regulation of Gq-coupled receptors’ signaling.

The pattern of agonist-dependent phosphorylation by GRKs has been studied in more detail for the M3 muscarinic receptor, a relative of the M1 receptor. The rat M3 possesses 53 Ser/Thr residues on the third cytosolic loop, clustered in six Ser/Thr-rich regions. Two of these clusters located in the N-terminal part of the loop, ^331^SSS^333^ and ^348^SASS^351^, are the key regions phosphorylated by GRK2: mutation to alanines of either of the clusters reduces phosphorylation, and mutation of both clusters reduces the phosphorylation level by 75% [95]. It was not directly determined in this study whether or how these mutations affected receptor internalization and/or arrestin recruitment. However, the M1 receptor with deletion of the ^289^Cys-His^330^ region, which partially covers the first cluster, was shown to be deficient in agonist-induced internalization. The authors attributed this effect to the loss of the Gβγ binding site, which they localized to this region. An earlier study identified ^286^ESLTSSE^292^ as critical for M3 internalization [96]. Although GRK2 appears to be the primary regulator of M1 receptor phosphorylation and desensitization, in vitro, the M3 receptor is readily phosphorylated by GRK3, better than by GRK2. It is also a good substrate for GRK5 and a substrate for GRK6 [52], contrary to earlier reports that found no role for GRK5/6 in phosphorylation of the M3 receptor [97]. Mice lacking GRK3 show enhanced airway contractile responsiveness to muscarinic stimulation [98], whereas mice lacking GRK5 did not differ from wild type [80]. The M3 receptor is the primary muscarinic receptor subtype regulating airway smooth muscle contraction [99,100]. Thus, GRK3 appears to regulate the M3 receptor function in vivo. Since the M3 receptor couples to Gq, its signaling, like that of the M1 receptor, could also be regulated by GRK2/3 in a phosphorylation-independent manner via the binding of the RGS homology domain of these GRKs to active Gαq, which participates in recruiting GRK2 to the plasma membrane, in addition to GRK2 interaction with Gβγ [101]. Arrestin recruitment, though, requires GRK2-dependent phosphorylation of the M3 receptor.

## 3. The Order of Phosphorylation: Sequential and Hierarchical?

Among other things, the large number of potential phosphorylation sites means that they are unlikely to be targeted all at once. Two models are conceivable: the sequence of phosphorylation of the available serines and threonines can be random, or they can be modified in a certain order. This issue, like many in GPCR regulation, was first explored in rhodopsin. Using a bright flash that activates ~40% of rhodopsin in the mouse retina (this is orders of magnitude higher than the light level where rods normally operate), it was shown that Ser343 is phosphorylated first, Ser338 is phosphorylated slower, and Ser334 even slower [102]. As dephosphorylation proceeds in the same order, the authors hypothesized that the order of phosphorylation reflects the accessibility of these residues, with those closer to the C-terminus being the most exposed and therefore hit first both by the kinase and by phosphatase [102]. While this study suggested that serines in the rhodopsin C-terminus are phosphorylated first, a recent study suggested that phospho-threonines are more important for arrestin-1 binding in photoreceptors [103]. Another in vivo study showed that replacing just one phosphorylatable residue, Ser343 or Ser338, with alanine prolongs photoresponse recovery, whereas replacing all three serines in the rhodopsin C-terminus with alanines, while leaving the threonines in place, prolongs it even more [104]. Notably, both studies present similar numbers: elimination of the three serines prolongs the recovery 2–3-fold [103,104]. In the crystal structure of the arrestin-1 complex with rhodopsin, the phosphates attached to Thr336 and Ser338 are visible and interact with two positive patches on the arrestin-1 surface, whereas the third patch is occupied by the negatively charged Glu341 [105]. Ser343 is not visible, and if we extrapolate the rhodopsin C-terminus, it would project away from arrestin-1 [105]. These contradictory data have yet to be reconciled. 

In contrast to many other protein kinases, none of the GRKs has a consensus phosphorylation site, i.e., a particular sequence context of serines and threonines they preferentially target. This makes biological sense, as the intracellular elements of GPCRs demonstrate very low sequence conservation [35]. However, it was shown using synthetic peptides that GRK2 has a preference for serines and threonines preceded by a negatively charged residue (which can be an upstream serine or threonine already phosphorylated), while GRK1 (rhodopsin kinase) prefers serines and threonines that have negatively charged residues downstream [106]. 

In some cases, the phosphorylation appears to be hierarchical, i.e., certain sites must be phosphorylated first, whereupon others can be modified. One interpretation of this phenomenon is that a receptor phosphorylated at certain sites becomes more “attractive” for some (or all) GRKs. However, an alternative explanation is that phosphorylation might simply induce conformational rearrangement of the intracellular receptor elements, which results in the exposure of some sites inaccessible in the unphosphorylated receptor. The latter explanation appears correct for D1 dopamine receptor phosphorylation, where the C-terminal sites are likely phosphorylated first to give the kinase access to the sites on the third loop [33]. A related important question is how GRK phosphorylation of specific residues impacts the receptor signaling and/or trafficking. Examples of this mode of GPCR phosphorylation, as well as the functional consequences, are discussed in more detail below for receptors for which there are sufficient data to yield a reasonably coherent picture.

### 3.1. Opioid Receptors 

Arguably, opioid receptors are the most clinically relevant peptide receptors. Synthetic ligands of these receptors are widely used to treat acute and chronic pain. Opioid agonists also cause addiction. These are the reasons that the function of opioid receptors has been studied more extensively than other neurotransmitter-binding GPCRs. Additionally, the opioid receptors happen to have a limited number of potential GRK phosphorylation sites, as compared to other important neurotransmitter/neuromodulator receptors such as muscarinic or dopamine (see above sections). This makes the work of identifying the sites and assessing their relative functional contribution in cultured cells and living animals feasible. Four opioid receptors (OR) have been cloned, the mu (MOR), kappa (KOR), delta (DOR), and nociceptin/orphanin FQ (NOR) receptors. Opioid receptors mostly couple to the pertussis toxin-sensitive Gi/o subfamily of G proteins and, to a lesser extent, to Gz (reviewed in [107]). While numerous protein kinases phosphorylate different types of opioid receptors, in most cases, one or more GRKs and one or both non-visual arrestins are involved in desensitization and internalization of ORs [107]. The regulation of NOR has not so far been extensively studied; most available data were obtained with MOR and DOR. The existing evidence suggests that all opioid receptors are regulated by GRKs (reviewed in [107,108]) and that opioid receptors fit the general paradigm of two-step homologous desensitization described above (Figure 1). The opioid receptors are a good subject for the discussion of GRK-dependent regulation for two reasons. First, they possess relatively few potential phosphorylation sites and apparently even fewer are actually phosphorylated and determine the receptor’s behavior. Second, the phosphorylation/desensitization of the opioid receptors, particularly MOR, is relatively well studied on account of the potential role of the desensitization mechanisms in the clinical efficacy and/or side effects of opioid drugs.

Each opioid receptor possesses multiple potential phosphorylation sites for GRKs. Limited information is available about the phosphorylation of opioid receptor subtypes induced by endogenous opioid peptides in the brain. However, the phosphorylation patterns induced by multiple synthetic opioid agonists and their functional consequences have been examined extensively, particularly for MOR. MOR has 11 serines and threonines, i.e., potential phosphorylation sites, in its C-terminus (Figure 3). GRK-dependent phosphorylation of MOR appears to be sequential and hierarchical, with Ser375 serving as the initial site [109]. All drugs induce phosphorylation at Ser375 in cultured cells [109,110,111] and in the brain [110,112]. Phosphorylation of only this residue, as seen upon application of morphine, results in weak arrestin recruitment that is not sufficient to drive substantial receptor internalization [111,113]. In contrast to morphine, synthetic high efficacy MOR agonists effectively drive MOR phosphorylation, arrestin recruitment, and receptor internalization [111,113]. For efficient arrestin recruitment and arrestin-mediated internalization, phosphorylation of multiple residues in the 10-residue C-terminal stretch ^370^TREHPSTANT^379^ was required. High-efficacy opioid drugs generally have the ability to cause phosphorylation of these extra residues, albeit to a different extent. DAMGO promotes phosphorylation not only of Ser375 but also of T370, T376, and T379 in its vicinity [109,110,114]. The synthetic opioids fentalyl and etonitazene also promoted MOR phosphorylation at multiple sites, including Ser375 and Thr370 [112,115]. Within the upstream stretch of phosphorylatable residues ^354^TSST^357^, Ser356 and T357 are phosphorylated in an agonist-dependent manner, but without any detectable effect on receptor internalization [113,114,116]. There is evidence that phosphorylation of this region enhances MOR-GRK2 and MOR-arrestin interactions, thereby facilitating phosphorylation of the key residues within the ^370^TREHPSTANT^379^ motif [113]. GRKs 2 and 3 seem to be responsible for phosphorylating most of the residues in MOR. Synthetic agonists promote GRK2/3-dependent phosphorylation of multiple residues [109,110]. Morphine, on the other hand, promotes Ser375 phosphorylation by GRK5 [110]. Morphine-induced phosphorylation of Ser375 was significantly enhanced by overexpression of GRK2 to the level of that caused by DAMGO, but the receptor internalization was only partially rescued [117]. These data demonstrate yet again that phosphorylation of other residues in addition to Ser375 is required for efficient receptor internalization. Elimination of the phosphorylation of all four residues in the ^370^TREHPSTANT^379^ stretch in the MOR C-terminus (Figure 3) was sufficient to block arrestin recruitment and MOR internalization [109,113,118]. In HEK cells with GRK2 or GRK3 deleted by CRISPR/Cas9, MOR phosphorylation and arrestin recruitment were reduced, particularly by the lack of GRK2. However, some residual arrestin-3 recruitment upon MOR activation remained [119]. In locus coeruleus neurons of mice lacking GRK3, MOR desensitization induced by Met-ENK was unaffected [120]. These data demonstrate the primary role of GRK2/3, likely mostly GRK2, in MOR phosphorylation but also suggest that other GRK(s) are able to take over in the absence of these kinases.

Since it is easier to detect, receptor internalization is often used as a readout for arrestin recruitment in lieu of receptor desensitization. However, these two processes are distinct and have differential mechanistic requirements, as we have seen above with other receptors. For example, phosphorylation of the four residues within the ^370^TREHPSTANT^379^ sequence is sufficient to support MOR internalization, whereas other residues contribute to MOR desensitization, since the mutation of all 11 C-terminal Ser/Thr residues is needed to abolish desensitization of MOR signaling via inwardly rectifying potassium channels GIRK [118]. Phosphorylation of the ^354^TSST^357^ sequence, together with that of the ^370^TREHPSTANT^379^ stretch, contributes to rapid MOR desensitization [121]. 

Much less is known about the pattern of GRK-dependent phosphorylation of other opioid receptors. DOR has seven potential phosphorylation sites in the C-terminus (Figure 3). Out of these, Thr358 and Ser363 are targeted by GRK2, with Ser363 being the primary site [123,124,125]. Phosphorylation of Thr361 after Ser363 by GRK2/3 has also been observed [126]. Strong DOR agonists such as DPDPE and DADLE (enkephalin analogues) induced robust DOR phosphorylation at both Ser363 and Thr361 and receptor internalization, whereas agonists that only caused Ser363 to be phosphorylated induced weak internalization [126]. KOR has only four potential C-terminal phosphorylation sites, and phosphorylation of one, Ser369 (in mouse KOR), by GRK3 is associated with arrestin recruitment and receptor internalization [127,128]. Later studies have demonstrated that all four residues are phosphorylated following stimulation with the selective agonist U50,488H, with Ser369 being the primary site that promotes phosphorylation of Thr363, as well as of the tandem Ser356/Thr357 [129]. The degree of multiple residue phosphorylation induced by an agonist correlated with the extent of KOR internalization [129]. Using the knockdown approach, it was shown that all four ubiquitously expressed GRK isoforms, GRKs 2, 3, 5, and 6, phosphorylate, albeit with different efficacy, all C-terminal serines and threonines of KOR [129]. NOR has six potential C-terminal phosphorylation sites (Figure 3). Ser363 appears to be the primary site phosphorylated by GRK3, but not GRK2, leading to the recruitment of arrestin-3 (but not arrestin-2). Mutation of this site to alanine was sufficient to abolish the receptor internalization [130]. Another study using phosphospecific antibodies found that four C-terminal residues (Ser346, Ser351, Thr362, Ser363) were phosphorylated by GRK2/3 sequentially, with Ser346 being phosphorylated first, followed by Ser351 and then Thr362/Ser363 [131]. GRK2/3 cooperate in phosphorylation of all four residues. NOR agonists showed differential propensity to activate G protein-mediated signaling and facilitate receptor phosphorylation [131]. In mice, agonists induce multisite phosphorylation and internalization in a dose-dependent and agonist-selective manner [131]. Thus, all opioid receptors followed the pattern of sequential hierarchical phosphorylation in the C-terminus that facilitated arrestin recruitment and receptor internalization. The degree of internalization appears to be roughly proportional to the extent of receptor phosphorylation at multiple residues, and in some cases, phosphorylation of a single residue caused by a weak agonist proved insufficient to recruit arrestin or induce receptor internalization. The relationship between receptor phosphorylation at various residues and receptor desensitization is harder to evaluate, because in most studies it has not been directly examined. It is possible that the requirements for receptor desensitization differ from those for internalization, as discussed above for MOR desensitization by morphine.

### 3.2. Cannabinoid Receptors 

Two cannabinoid receptors, CB1 and CB2, have been cloned. The first was CB1, the receptor protein for Δ9-THC, the major psychoactive constituent in cannabis, although now it is accepted that both CB receptors bind endogenous ligands, the endocannabinoids 2-arachidonoylglycerol (2-AG) and anandamide (AEA) (reviewed in [132]). CB1 is highly expressed in neurons, where it predominantly couples to the Gi/o subfamily of G proteins (although its coupling to Gs and Gq/11 has also been reported). Structurally, CB1 receptor resembles β2AR, with numerous potential GRK phosphorylation sites in the C-terminus (reviewed in [132]) (Figure 3). It appears that the phosphorylation of distinct parts of the CB1 C-terminus containing clusters of Ser/Thr residues phosphorylated by GRKs is involved in the receptor desensitization and internalization.

The removal of the last 14 residues of the rat CB1 receptor (residues 460–473) prevented its internalization in AtT20 cells [122]. However, CB1, with only 10 terminal residues removed, which left Thr461/Ser463 intact, was capable of internalization [122], suggesting an important role of these residues in the process. In contrast to AtT20 cells, CB1, truncated at residue 460, recruited arrestins and internalized normally in HEK293 cells [133]. Nevertheless, the C-terminal residues play an important role in the regulation of CB1 internalization, since mutation of four or all six distal C-terminal phosphorylation sites (Thr461A–Thr466A and Thr461A–Ser469A) precludes arrestin recruitment and internalization [133]. These data suggest that the proximal C-terminal residues can support arrestin recruitment and internalization but are normally masked by the distal part of the C-terminus. Studies in Xenopus oocytes demonstrated that desensitization of the CB1 receptor occurs in a GRK- and arrestin-dependent manner [134]. The upstream pair Ser426/Ser430 (Ser425/429 in human) is critical for CB1 desensitization, as double mutation Ser426A/Ser430A attenuates it [134,135]. Mutant Ser426A/Ser430A CB1 receptor can recruit arrestin, which leads to receptor internalization, but not desensitization [134,135]. Knockdown of GRK3 significantly increased ERK activation by wild-type CB1 receptor, but not the Ser426A/Ser430A mutant [136]. These data suggest that GRK3 phosphorylates these residues, so that the absence of GRK3 leads to diminished desensitization of CB1 and, consequently, to enhanced G protein-mediated ERK activation. Knockdown of GRKs 4, 5, and 6 did not alter ERK activation via wild-type CB1 by WIN55,212–2 [136]. These data suggest that GRK3-mediated phosphorylation of Ser426/Ser430 in CB1 receptors controls the receptor desensitization, whereas phosphorylation, presumably by GRK2/3, of C-terminal residues Thr461A–Ser469A mediates arrestin-dependent receptor internalization.

### 3.3. Class A GPCR Oligomers

There are numerous reports of oligomerization of class A (rhodopsin-like) GPCRs (e.g., see [137,138] and references therein). However, available evidence of the role of GPCR oligomerization in GRK/arrestin-dependent regulation of receptor function tends to be inconclusive, and all existing structures of GPCR complexes with potential signal transducers, G proteins, GRKs, and arrestins reveal 1:1 interaction ([139] and references therein); there is not enough experimental evidence to discuss this interesting subject meaningfully.

## 4. Barcode Hypothesis

Most GPCRs have a large number of serines and threonines on their intracellular elements that can potentially be phosphorylated by GRKs (Figure 2). This raises the possibility that different phosphorylation patterns of the same receptor might emerge, possibly via phosphorylation of active receptors by different GRKs, and that these patterns might result in different sets of events at the molecular level and, consequently, distinct biological outcomes. This idea was first expressed more than a decade ago [140] and was later aptly termed the “barcode hypothesis” [141]. 

Convincing data of the receptor “barcoding” by different GRKs come from the studies of chemokine receptors, many of which interact with more than one endogenous agonist. The chemokine receptors activated by different agonists may engage distinct GRK isoforms, often from the GRK2/3 or GRK4/5/6 subfamilies. Phosphorylation by different GRKs leads to diverse functional consequences and may or may not result in arrestin recruitment, desensitization, internalization, and/or arrestin-mediated signaling [142,143]. In other cases, receptors responding to unique agonists are phosphorylated by different GRKs at distinct sites, thus establishing a “barcode”. There is limited indirect evidence suggesting that GPCRs are indeed differentially barcoded by distinct GRKs. Phosphorylation of angiotensin II [144] and vasopressin V2 receptors [145] by GRKs 2 and 3 on the one hand and GRKs 5 and 6 on the other, presumably targeting different sites, results in arrestin binding that promotes internalization or ERK1/2 activation, respectively. Similar results were obtained with β2AR [146], suggesting that the identity of the GRKs that phosphorylate the active receptor determines the biological consequences of subsequent arrestin binding. In the same vein, a recent study of D1 receptor mutants suggested that the functional outcome of arrestin binding to differentially phosphorylated D1 receptors might be different, favoring Src or ERK1/2 activation in case of phosphorylation at different sites [147].

The next question that must be asked is what is the functional significance of the phosphorylation of each site and their possible combinations? Active GPCRs interact with at least three protein families, G proteins, GRKs, and arrestins [148], each of which has several members. Usually, the effect of differential phosphorylation is considered in terms of different modes of arrestin binding, without even taking into account that cells have two non-visual arrestins with distinct functional capabilities. Conceivably, phosphorylation of particular sites can change receptor preference for individual G proteins or for GRK isoforms phosphorylating the remaining sites, as well as for one of the two arrestin isoforms. Unfortunately, these questions are virtually never asked experimentally. For most GPCRs, no comparison of the relative ability of unphosphorylated GPCRs and the same receptors phosphorylated at certain sites to couple to different G proteins, recruit different GRKs, or recruit either of the two non-visual arrestins was made in experiments with purified proteins or even in cultured cells. As the experimental approaches necessary to test this are complicated, the barcoding of GPCRs by GRKs remains, on the whole, largely unexplored.

## 5. Agonist Dependence of GRK Action

In the classical paradigm, GRKs only phosphorylate agonist-activated GPCRs. However, there are fragmentary data suggesting that this is not always the case. A study with purified neurotensin receptor 1 (NTSR1) reconstituted into nanodiscs showed that GRKs 2 and 5 differentially phosphorylate it. The action of GRK2 was strictly agonist-dependent, whereas GRK5 phosphorylated NTSR1 independently of its activation. GRK2 phosphorylated only C-terminal Ser residues, whereas GRK5 phosphorylated Ser and Thr residues in both the third intracellular loop and the C-terminus [149]. It appears that in the case of NTSR1, GRK2 does not require acidic residues upstream of the phospho-acceptors (in contrast to what was found for β2AR and MOR) [149]. Another study of the activity of all four ubiquitously expressed GRKs (GRKs 2, 3, 5, and 6) was performed in cultured cells. While GRK2 and GRK3 phosphorylated co-expressed β2AR and M2R in a strictly activation-dependent manner, GRK5 and GRK6 demonstrated significant phosphorylation of these receptors in the absence of agonists [52]. For β2AR, the same difference between purified GRK2 and GRK5 was observed, with purified receptor reconstituted into nanodiscs [52]. GRK5 also effectively phosphorylated opsin (rhodopsin devoid of retinal) and even dark rhodopsin with covalently bound inverse agonist 11-cis-retinal. Interestingly, all four GRKs selectively phosphorylated active dopamine D1R, i.e., in the presence of an agonist, but not in its absence [52].

In the case of rhodopsin [13], β2AR [14], and D2R [15], it was shown that GRKs bind GPCRs and are activated by direct interaction with the receptor. In view of this, one possible explanation for the phosphorylation of inactive receptors is that GRKs bind them and force them into an active-like conformation that in turn activates GRKs. This could explain why the same GRK5 phosphorylates some inactive GPCRs, while phosphorylating others in a strictly activation-dependent manner [52]: some receptors might be more flexible than others, so the efficacy of this binding-induced fit and kinase activation can differ widely depending on the receptor. The reported much higher phosphorylation by GRK5 of conformationally loose ligand-free opsin than of 11-ci-retinal liganded dark inactive rhodopsin [52] supports this model. 

## 6. From Neurotransmitter Receptor Regulation to Neural Adaptation: The Role of GRKs

When discussing GRK-dependent regulation of the neurotransmitter receptors, yet another set of questions must be asked: whether or how distinct molecular events affect neural function. It is not difficult to imagine that altered availability and/or function of GRKs, leading to impaired receptor desensitization/trafficking, would result in enhanced or reduced responsiveness to agonists, as in cultured cells. However, chronic treatment with neurotropic drugs often results in long-term neural plasticity, altering the responsiveness to these drugs, which develops over hours or days, in contrast to the much faster timescale of receptor phosphorylation and arrestin recruitment, both of which occur within minutes in cultured cells. This long-lasting plasticity is initiated by agonist stimulation of the neurotransmitter receptors, which would initially engage GRKs, and it is a legitimate—and intriguing—question whether GRK-dependent phosphorylation of the neurotransmitter receptors plays a role in the development of long-term neural plasticity.

### 6.1. GRKs in the Regulation of Acute Responsiveness to Neural Stimulation 

Studies in mice lacking individual GRK isoforms have demonstrated supersensitivity to acute stimulation of select neurotransmitter receptors. The loss of GRK6 confers supersensitivity to dopaminergic stimulation [63]. The dopaminergic supersensitivity at the behavioral and signaling levels associated with L-DOPA-induced dyskinesia can be counteracted by overexpression of GRK6 [150,151]. Based on these data, it has been concluded that GRK6 is the primary kinase regulating the dopamine receptors in vivo [81]. Global knockout of GRK2 is embryonically lethal [81]. Selective deletion of GRK2 in the D1-expressing neurons resulted in behavioral and neurochemical supersensitivity to psychostimulant dopaminergic drugs such as cocaine [152]. In contrast, in mice with GRK2 deleted in neurons expressing adenosine A2A receptors, i.e., mostly striatal neurons harboring postsynaptic dopamine D2 receptors, the behavioral sensitivity to cocaine was normal [152]. These data suggest that GRK2 is mostly involved in the regulation of the acute responsiveness of D1, but not D2, receptors. They are also in agreement with the in vitro results showing that GRK phosphorylation is dispensable for the D2 receptor regulation. Deletion of GRK2 in neurons expressing D2 receptors, though, resulted in enhanced spontaneous locomotion and reduced behavioral and neurochemical sensitivity to cocaine [152], which appears counterintuitive. However, one must remember that D2 autoreceptors regulate the release of dopamine at dopaminergic terminals. In the case of GRK2 deletion, it turned out that dopamine release, both tonic and phasic drug-evoked, was inhibited, likely due to hyperactive D2 autoreceptors, and this effect gave rise to blunted responsiveness to cocaine. This is a good illustration of how the regulatory effect of GRK-mediated phosphorylation of neurotransmitter receptors depends on the brain circuitry, i.e., the specific location and/or functional role within the circuitry of the cells bearing the receptors in question. This notion is also well illustrated by the fact that mice with selective deletion of GRK2 in cholinergic neurons show no changes in the dopaminergic functions [153]. To date, no experiments with cell-specific deletion of GRK6 have been performed to determine whether the striatal D1 or D2 receptor is the primary target.

Loss of GRK5 results in supersensitivity to muscarinic M2 receptor stimulation [77,80]. GRK5 deficiency impairs desensitization of M2/M4 autoreceptors, causing inhibition of the hippocampal acetylcholine release and cholinergic hypofunction [154]. The acetylcholine deficiency caused by the loss of one copy of the GRK5 gene in mice overexpressing β-amyloid precursor protein (APP) with the Swedish mutation (Tg2576) exacerbates the accumulation of β-amyloid [155]. Selective deletion of GRK2 in the brain cholinergic neurons caused reduced sensitivity to the effects of the muscarinic agonist oxotremorine such as hypothermia, hypolocomotion, salivation, and antinociception [153]. The hypothermia is mediated by M2 receptors, hypolocomotion is caused by the action via M1 and M4 receptors, and salivation is governed by M1, M3, and M4 receptors [78,79,156,157]. Since the GRK2 loss occurred in cholinergic neurons, it is likely to primarily affect muscarinic autoreceptors such as M2/M4. Unfortunately, the location within the brain circuitry remains unknown, which makes it impossible to interpret these data at the molecular level, since there are many cholinergic cell groups with diverse functions. Collectively, these studies have highlighted an important role of GRKs in the regulation of receptor responsiveness to stimulation. The functional role of GRK-mediated phosphorylation of specific receptors at precisely defined sites within the brain circuitry has not yet been addressed.

The exact relation between GRK-mediated receptor phosphorylation and in vivo adaptations to receptor stimulation has been studied in some detail in the case of opioid receptors. Physiologically, opioids induce analgesia in humans and animals, and classic opioid drugs exert their effects via MOR. If opioid analgesia is mediated by G protein activation via MOR, desensitization of the receptor would serve to limit the extent and duration of the analgesic response. Indeed, knockin mice expressing MOR with Ser 375 mutated to alanine (Ser375Ala), which cannot be phosphorylated, showed greater antinociceptive response to morphine and fentanyl [112]. In knockin mice with multiple serines and threonines in MOR mutated to alanines, which made the receptor increasingly unable to be phosphorylated, recruit arrestins, and be desensitized, antinociceptive responses to morphine and fentanyl were significantly enhanced and prolonged [158]. Mice lacking GRK3 showed unchanged antinociceptive responses [115,159], presumably because GRK2 was still available to phosphorylate MOR. Unfortunately, the role of GRK2 has not so far been examined using this approach, because global GRK2 knockout in mice is embryonically lethal. Interestingly, the loss of GRK5 had the opposite effect: GRK5 knockout mice demonstrated diminished antinociception as compared to wild type [115]. This latter result is counterintuitive, suggesting that some action(s) of GRK5 is involved other than its role in receptor phosphorylation/desensitization. Similar to the opioid receptors, impaired desensitization of cannabinoid receptors results in enhanced acute responsiveness to stimulation. In mice with two prime phosphorylation sites (Ser426 and Ser430) in the C-terminus of the CB1 receptor mutated to alanines, acute sensitivity to Δ9-THC was increased. Desensitization and downregulation of CB1 in the spinal cord was absent. In autaptic cultured hippocampal neurons, endocannabinoid responses were enhanced and their desensitization reduced [160]. 

### 6.2. GRK-Mediated Rapid Desensitization in Long-Term Neural Adaptations 

GRK-arrestin-mediated homologous desensitization of MOR is a short-term adaptation that eventually results in re-sensitization or a longer-lasting adaptation such as receptor degradation/downregulation. Treatment with opioid drugs leads to a loss of responsiveness referred to as tolerance, which could be short-term (acute tolerance) or long-term (chronic tolerance). From a physiological perspective, it is important to understand the relationship between the opioid receptor desensitization and tolerance. This subject has been extensively reviewed previously (e.g., see [161]). An interesting question is whether GRK-dependent receptor phosphorylation followed by arrestin recruitment and desensitization and/or internalization plays any role in the long-term neural adaptations, such as tolerance, caused by the use of opioids. Studies in mice carrying MOR with mutated phosphorylation sites have contributed important information. Knockin mice with MOR mutant Ser375Ala, the site critical for the initiation of the phosphorylation cascade, showed greatly diminished tolerance to fentanyl, etonitazene, or DAMGO [112]. Interestingly, tolerance to morphine did not change [112]. Long-term tolerance to fentanyl, but not morphine, was completely abolished in mice expressing the Ser375Ala MOR, which was accompanied by reduced desensitization of the locus coeruleus neurons [158]. Interestingly, tolerance for morphine was also greatly diminished in mice expressing MOR mutants lacking other phosphorylation sites in addition to Ser375, suggesting that phosphorylation of multiple sites contributes to morphine tolerance. However, morphine tolerance was not completely eliminated, even in mice lacking all 11 phosphorylation sites, suggesting that mechanisms independent of GRKs/arrestins are involved specifically in morphine tolerance [158]. Interestingly, withdrawal symptoms were unchanged in mice expressing phosphorylation-deficient MOR mutants [158], indicating that MOR desensitization involving GRK phosphorylation and arrestin recruitment does not underlie withdrawal symptoms. Acute tolerance to fentanyl, methadone, and oxycodone was reduced in GRK3 knockout mice [159,162], whereas tolerance to morphine was unchanged in mice lacking GRK3 or GRK5 [115]. Mice lacking GRK5, but not GRK3, failed to form conditional place preference (CPP) to morphine [115], which is indicative of a reduced rewarding effect of morphine in these animals. Similar to GRK3 knockout mice, mice expressing mutant Ser375Ala MOR formed strong CPP to morphine [115], suggesting that GRK5-dependent phosphorylation of this residue is not required for morphine CPP. The authors suggested a role of agonist-induced activation of ERK1/2, although the mechanism of ERK involvement and its activation in this paradigm remains unclear.

DOR agonists are efficacious in chronic pain but, similarly to MOR agonists, induce tolerance upon prolonged use. The role of GRK-dependent phosphorylation in tolerance to DOR agonists remains essentially unexplored. However, there is evidence that tolerance depends on the availability of arrestin-2. Thus, loss of arrestin-2 reduced tolerance to the DOR agonist SNC80, which induces strong DOR internalization, but not to ARM390, which does not [163]. DOR phosphorylation was not examined in this study, but since DOR internalization correlates with phosphorylation of Thr361 in addition to Ser363, these agonists might differ in their ability to cause DOR phosphorylation. Lack of KOR phosphorylation at the key Ser369 residues in mice lacking GRK3 resulted in significantly reduced chronic antinociceptive tolerance to the KOR agonist U50,488 without affecting the acute analgesic effect [128]. 

Generally speaking, chronic tolerance appears to involve GRK-dependent phosphorylation of opioid receptors enabling arrestin recruitment, and when these functions are compromised, tolerance is diminished. The mechanism of this effect is intriguing, since the timings of these events—GRK/arrestin-dependent homologous desensitization and tolerance—are so different (discussed here [161]). However, the data do indicate the requirement for GRK-dependent receptor phosphorylation in the development of tolerance. Speaking specifically about MOR, tolerance to high-efficacy MOR agonists seems to depend on MOR phosphorylation at Ser375, which is the key residue for the initiation of the phosphorylation cascade leading to MOR desensitization/internalization. However, tolerance to morphine seems to involve additional mechanisms, although phosphorylation of multiple MOR residues appears to be a contributing factor.

In mice with the Ser426 and Ser430 phosphorylation sites in the C-terminus of the CB1 receptor mutated to alanines, dependence on Δ9-THC was increased, but tolerance delayed, which was accompanied by enhanced acute responses [160]. Tolerance to the antinociceptive effect of WIN55,212–2, another synthetic cannabinoid agonist, was also delayed in these mice [164]. Thus, in the case of cannabinoid receptors, tolerance appears to involve GRK phosphorylation and likely subsequent arrestin recruitment, as in opioid receptors. However, the time course of physiological tolerance to opioids and cannabinoids (days to weeks) is much longer than of receptor phosphorylation and subsequent arrestin binding (minutes) or even receptor internalization and downregulation (hours). Thus, it appears that receptor phosphorylation and/or subsequent formation of the receptor–arrestin complex initiates longer-term regulatory processes in neurons that are not apparent in cell culture models. 

### 6.3. Neurotropic Drugs: To Bias or Not to Bias? 

All therapeutically active drugs have side effects. Since the discovery of arrestin-mediated signaling by GPCRs, the idea of biased signaling, i.e., signaling engaging only one branch of the GPCR pathway, either G protein or arrestin, has gained popularity. For the biased signaling to be possible, the two branches, the G protein- and arrestin-mediated, should be independent, and originally they were believed to be. There are obvious limitations to this independence. The fact that GRKs 2 and 3 use two products of G protein activation, Gβγ [60,61,62] and activated Gαq/11 [101], for membrane localization suggests that G protein activation likely plays a role in GPCR phosphorylation by these GRKs, which is often necessary for arrestin binding. This might limit what arrestins can do in the absence of G protein activation. On the other hand, GRKs 4/5/6 have C-terminal lipid modifications and/or specific sequences mediating their attachment to the membrane independently of G proteins (reviewed in [5]). Thus, phosphorylation of the same GPCRs by GRKs 5/6 instead of GRKs 2/3 might bypass the need for G protein and enable arrestin recruitment and arrestin-mediated signaling. Recently, a question arose as to whether G protein activity is required for arrestin-mediated signaling in some capacity other than for the GRK recruitment to the receptor. It has been reported that the presence of functional G proteins is indispensable for arrestin-mediated activation of the ERK pathway [165]. In cells, the ERK pathway is activated via G protein as well as arrestin-dependent mechanisms [166,167], but it can be activated exclusively via G proteins independently of arrestins [168]. Whether the reverse is true has not yet been unambiguously determined (see short discussion in [169]). The issue of the potential and limitations of G protein vs. arrestin bias of GPCR ligands was recently discussed in depth [148].

On the physiological side, the idea is based on the notion that the therapeutic action might be mediated by one signaling branch and the side effects by another. The attraction of this theory is obvious but the experimental basis is limited (discussed in [148]). Arguably, nowhere has this theory received more attention than in the field of opioid therapy. Opioid drugs remain the mainstay of pain management. Their utility is limited, however, not only by the tolerance that develops upon long-term use but also by multiple unwanted side effects including life-threatening respiratory depression, gastrointestinal disturbances, and addiction. Since GPCRs signal via both G protein- and arrestin-mediated pathways, it has been suggested that the therapeutic antinociceptive effects of opioid drugs are mediated by G proteins, whereas arrestin-dependent signaling is responsible for the side effects. This notion is based on studies of G protein-biased opioid agonists such as PZM21 [170] and TRV-130 [171]. TRV130 induced less MOR phosphorylation and arrestin recruitment than unbiased agonists. It is a potent analgesic with less evident gastrointestinal dysfunction and respiratory suppression than morphine [171,172]. However, direct proof that the arrestin-mediated signaling that takes place following MOR phosphorylation and arrestin recruitment is responsible for the opioid side effects is lacking. It is important to note that in many cases—clearly in the case of opioid receptors, for which phosphorylation by GRKs is a prerequisite for arrestin recruitment—bias for or away from arrestin is in reality a bias for or away from specific GRKs. What hampers the meaningful discussion of biased signaling is the lack of information on the phosphorylation patterns of the most relevant receptors and their potential barcoding by GRK isoforms. A recent important study using knockin mice expressing phosphorylation-deficient MOR with all or nearly all phosphorylation sites eliminated demonstrated that the side effects of both morphine and fentanyl were, if anything, exacerbated [158]. This effect was accompanied by enhanced analgesia and diminished tolerance. Furthermore, the EC_50_ values for morphine analgesia and respiratory depression/constipation across lines with different phosphorylation levels are highly correlated (R^2^ > 0.9), suggesting that these effects are not independent. These experiments strongly suggest that all major physiological effects of MOR activation, both good and bad from our perspective, are largely mediated by G proteins.

## 7. Conclusions

Existing data show that most GPCRs, including neurotransmitter receptors, are regulated by GRKs. However, the relative role of the five distinct non-visual GRKs expressed in all vertebrates in the regulation of these receptors has not been experimentally established. The situation is further complicated by the possibility that the role of GRK subtypes in the regulation of the same receptor is not necessarily the same in different neurons, possibly even under different conditions in the same neuron. In most cases, GRK phosphorylation promotes the binding of one or both non-visual arrestins. The two ubiquitous non-visual arrestins are functionally distinct, as their duality persisted in vertebrate evolution for millions of years [24]. There is no reason to assume that they are interchangeable. Hence, the role of each subtype in each type of neuron must be elucidated.

The generalized paradigm of GPCR desensitization (Figure 1) was developed using rhodopsin as a model GPCR. However, rhodopsin in vertebrates does not internalize: it is localized on intracellular discs in rods, which are topologically equivalent to endosomes containing internalized receptors. While desensitization and internalization are often used interchangeably, these are two different processes, as was clearly demonstrated by experiments with M2 and D2 receptors. Arrestin binding to the phosphorylated receptor can play a role in both, as in β2AR, or in desensitization but not internalization, as in the case of M2 receptors, or in internalization but not desensitization, as in the case of D2 receptors. Therefore, the role of phosphorylation in each process needs to be tested separately, which is rarely done. Moreover, phosphorylation at any site can change receptor preference for a particular class of G proteins and/or for a particular GRK. Neither of these effects is routinely tested.

Experimental testing of the role of individual phosphorylation sites by mutagenesis has serious caveats that must be kept in mind. Some GPCRs possess a huge number of potential phosphorylation sites, which makes the task of determining the key residues for GRK phosphorylation truly daunting. The replacement of Ser/Thr with alanines or valines to determine the sites phosphorylated by GRKs is a common approach. However, replacement of a serine with an alanine, in addition to preventing phosphorylation, changes the H-bonding capability of the residue. Replacement of threonine with alanine or valine also changes the H-bonding capability, as well as the size of the side chain. Thus, it is not always apparent whether these serines and/or threonines participate in arrestin binding as such or whether their phosphorylation by GRKs is needed for arrestin recruitment. Often, the “phosphomimetics” aspartate or glutamate are used in place of putative phosphorylation sites to determine whether phosphorylation is required. However, it has been experimentally shown that these “phosphomimetics” do not always mimic phosphorylated serine or threonine, likely because both the size and the charge at physiological near-neutral pH is quite different. Thus, if alanine substitution and knockout of a particular GRK yields the same result, while replacement of the residue with a phosphomimetic rescues the GRK knockout phenotype, we can conclude that the observed changes were due to lack of phosphorylation. If each change produces a unique phenotype and there is no observable rescue, we cannot conclude anything with confidence. We should keep in mind that the results of only one or even two of these three approaches are also inconclusive. The full battery of tests is almost never performed, even in cultured cells. Conclusions based on incomplete experimental testing should be always taken with a grain of salt. Unfortunately, collecting such complete sets of data in vivo is rarely feasible. However, as demonstrated with the mouse MOR knockin experiments [158], even a limited approach can be quite illuminating.

A recent structural study suggests that GPCRs must have a certain “phosphorylation code”, i.e., a particular spacing between phosphorylated and/or negatively charged residues, in order to bind arrestins with high affinity [105]. Two codes were proposed: PxPxxP/E/D and PxxPxxP/E/D, where P is phosphorylated serine or threonine, E is glutamic acid, D is aspartic acid, and x stands for any amino acid residue [105]. These codes contain three negative charges, matching three positively charged patches on the receptor-binding surface of arrestins. This is consistent with earlier findings that three rhodopsin-attached phosphates are necessary for tight binding and rapid quenching of signaling both in vivo [104] and in vitro [173]. Interestingly, β2AR, which was the first non-visual GPCR shown to bind arrestin, does not have a complete phosphorylation code [105]. It should be noted that none of the opioid receptors has either of these complete codes (Figure 3), yet they do bind non-visual arrestins. Thus, it appears that incomplete codes and/or phosphorylated residues with different spacing can do the job.

The connection between fairly rapid (minutes to hours) molecular events involving GRKs and arrestins, which were mostly established in vitro and in cultured cells, and physiological mechanisms generating long-term (days to weeks) changes in vivo is one of the mysteries that needs to be resolved. This is further complicated by the known fact that behavioral changes involve circuit effects that cannot be reproduced in cell culture. Biologically relevant in vivo consequences of phosphorylation of particular residues in each GPCR can only be established using genetically modified animals. While these experiments take a lot of time, effort, and resources, they are necessary, as extrapolation from test tube and cell culture tends to be misleading. 

## Figures and Tables

**Figure 1 cells-10-00052-f001:**
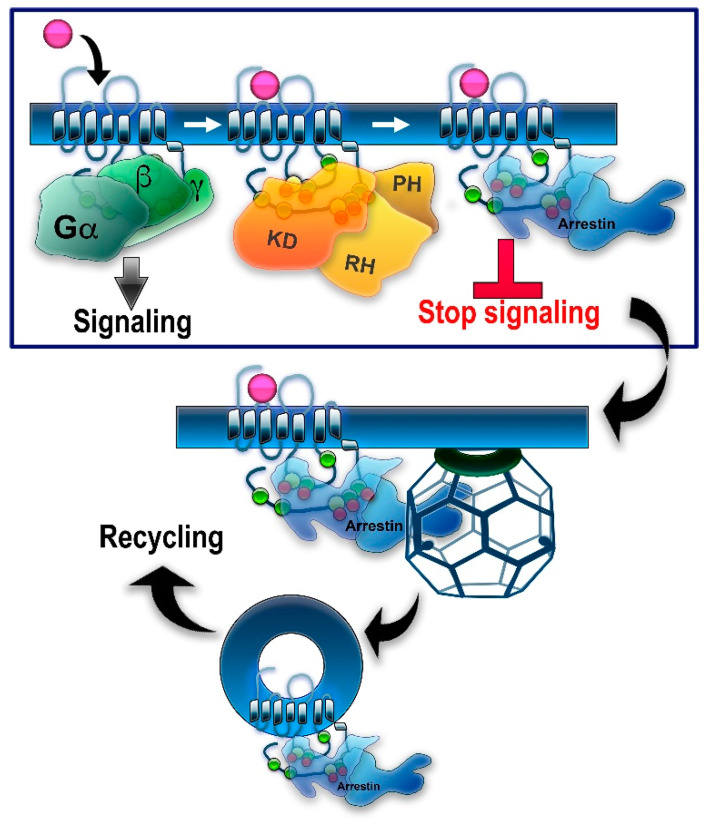
GRK/arrestin-mediated homologous desensitization of GPCRs. The classical paradigm of homologous GPCR desensitization posits that active receptors are phosphorylated by one or more GRKs, whereupon arrestins selectively bind to active phosphorylated receptors. Relevant phosphorylation sites in different GPCRs are localized in the C-terminus, 3rd cytoplasmic loop, and/or other cytoplasmic elements of the receptor (see Figure 2). Putative phosphorylation sites are shown here as circles: serines—green; threonines—yellow. Bound arrestin shields the cytoplasmic part of the receptor, precluding further G protein activation—this constitutes receptor desensitization (boxed). Arrestin binding to a GPCR induces the release of the arrestin C-tail. C-tails of non-visual arrestin-2 and -3 carry binding sites for the main components of the internalization machinery of the coated pit, clathrin, and clathrin adaptor AP2 (clathrin cage is shown). Arrestin binding promotes GPCR internalization via clathrin coated pits followed by receptor resensitization/recycling or degradation/downregulation. Note that all GRKs have a kinase domain (KD) and an RGS homology domain (RH), but only GRK2/3 have a pleckstrin homology domain (PH) that binds Gβγ, whereas other GRKs use different mechanisms of membrane localization.

**Figure 2 cells-10-00052-f002:**
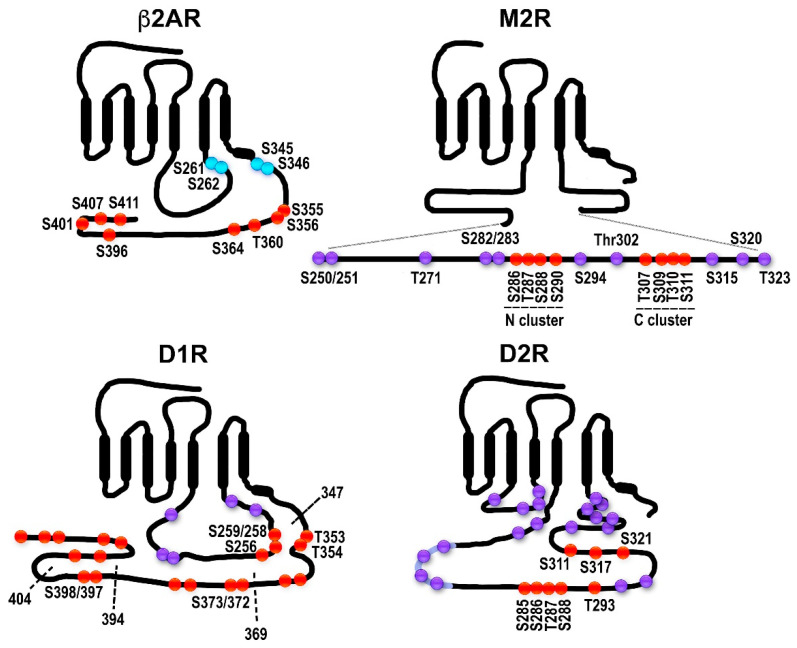
Distribution of potential and established GRK phosphorylation sites in selected GPCRs. The classical model does not necessarily apply to all GPCRs: in some cases, receptor phosphorylation is not necessary or plays a minor role in arrestin binding; some arrestin-associated receptors do not internalize via coated pits; for others, arrestins appear to mediate receptor internalization, but not desensitization, etc. β2-adrenergic receptor (β2AR). The sites in the human β2AR are shown. The sites of PKA phosphorylation are shown in blue. The sites of GRK phosphorylation are shown in red. All GRK targets are localized in the C-terminus, as in rhodopsin, β1AR, opioid and cannabinoid receptors, and many other GPCRs. Muscarinic M2 receptor (M2R). The sequence and GRK phosphorylation sites in the human M2R are shown. All putative sites are located in the 3rd cytoplasmic loop; the actual phosphorylation sites have been localized to the central part of the 3rd loop (Ser250-Thr323 shown here as an insert). The sites in the two characterized clusters are shown in red, the other sites in magenta. Two clusters of phosphorylatable residues (red) with different functions were described. Thr307-Ser311 (C cluster) appears to be necessary for desensitization; when the N cluster (residues Ser286-Ser290) is mutated to alanines, the receptor still desensitizes. Arrestin binding depends on the C-cluster [30]. Internalization is promoted by phosphorylation of either cluster [30,31,32]. Other potential phosphorylation sites within the region are shown in purple. Dopamine D1 receptor (D1R). D1R has sites in both the C-terminus and the 3rd cytoplasmic loop. The rat D1R is shown. The sites labeled in red have been shown to be phosphorylated in an agonist-dependent manner either in truncation experiments or via mutations to alanines [33]. Truncations used in [33] are shown as dotted lines with the last residue remaining labeled. Other potential phosphorylation sites are shown in purple. Dopamine D2 receptor (D2R). The rat D2R is shown. All phosphorylatable sites are in the 3rd cytoplasmic loop. Eight sites phosphorylated in an agonist-dependent manner by GRKs are shown in red [34]. Some of the other potential phosphorylation sites are shown in purple.

**Figure 3 cells-10-00052-f003:**
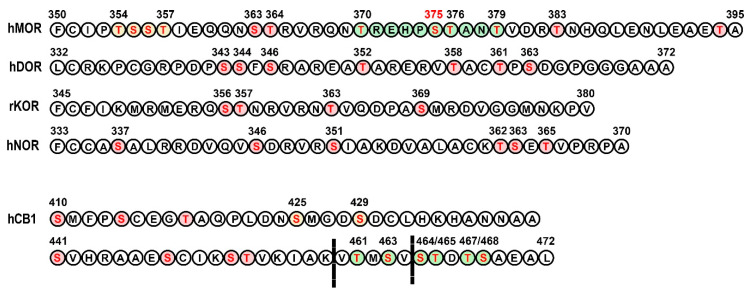
GRK phosphorylation sites in opioid and cannabinoid receptors. Top: C-termini of the four subtypes of human opioid receptors, with residue numbers indicated above. The phosphorylation of MOR was shown to be hierarchical, with Ser375 phosphorylated first, whereupon other residues within the stretch 370TREHPSTANT379 (highlighted in green) can be phosphorylated. Phosphorylation of another upstream cluster, 354TSST357 (highlighted in yellow), contributed to desensitization. Other phosphorylatable Ser and Thr are also shown (highlighted in pink). Bottom: The C-terminus of the human cannabinoid receptor CB1. Critical phosphorylation residues in the distal C-terminus are highlighted in green; the lines show the position of truncation of 10 or 14 terminal residues described in [122]. Two residues playing a key role in receptor desensitization, Ser425/Ser429 (Ser426/Ser430 in the rat), are highlighted in yellow; other phosphorylatable residues in pink. All putative phosphorylation sites are fully conserved in the human, rat, and mouse CB1.

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
