# Peer review of "GRKs as Modulators of Neurotransmitter Receptors"

_cells, 2020, doi:10.3390/cells10010052_

Round 1
Reviewer 1 Report
The article entitled ”GRKs as modulators of neurotransmitter receptors” by E.V. Gurevich and V.V. Gurevich is an extensive review on GRKs in neurotransmission. The article is well written and understandable by non-specialists.
However it may be of an easier access if the sections between line 521 to 787 could be condensed to the main message. The conclusion is also too long.
The illustration could be improved:
Figure 2: The font size is too small, I, particular the phosphorylation sites.
Figure 3: In the legend it is indicated that yellow and green sites have special functions but they are not readable even when the figure is magnified.
Minor typing error could be corrected line 830 vrsual, this reference is not accessible through pubmed.
Author Response
Reviewer #1
However, it may be of an easier access if the sections between line 521 to 787 could be condensed to the main message. The conclusion is also too long.
We shortened indicated part, but we do not believe that the text between lines 521 and 787 can be condensed to one main message, as it contains several distinct messages:
- Hierarchical phosphorylation of cannabinoid receptors.
- Possible barcoding of GPCRs by GRKs.
- Dependence of GPCR phosphorylation by GRKs on receptor activation (including possible explanation of the mechanism of the phosphorylation of inactive GPCRs by GRKs).
- The relationship between rapid GPCR phosphorylation followed by arrestin recruitment and long-term neuronal plasticity responsible for tolerance and behavioral changes.
We also shortened conclusions as much as possible to leave several messages contained therein clear.
The illustration could be improved:
Figure 2: The font size is too small, I, particular the phosphorylation sites.
Figure 3: In the legend it is indicated that yellow and green sites have special functions but they are not readable even when the figure is magnified.
Thank you! We modified figures, as requested.
Minor typing error could be corrected line 830 vrsual, this reference is not accessible through pubmed.
Thanks! Typo corrected. Indeed, this reference is not accessible via PubMed, likely because it is ~40 years old and the paper was not in English. However, we believe that it is important to retain this reference, as this is the first report of rhodopsin sequence.
Reviewer 2 Report
This is a well written and timely review. GRK signalling comes into and out of vogue by seems to me to be a central feature of GPCR biology. There is still a great deal we don't understand and that comes out in this comprehensive review.
One issue they don't cover is the idea that a further point of signal integration mediated by GRKs might be reflected in receptor dimers and oligomers. How do things like barcodes in phosphorylation patterns get integrated into responses from GPCR heteromers? No one has seriously examined this issue- at least they could say that.
The authors could also differentiate between a role for G proteins in recruiting GRKs and beta-arrestin- i.e. a structural role in GPCR signalling complexes and the need for activation of the G protein per se as in their discussion of biased agonists and the D2DR.
Minor Issues
There is some debate regarding whether or not PKA causes the beta2-AR to switch from Gs to Gi. The authors should nuance this to reflect the debate.
Xenopus is misspelled on occasion.
Author Response
Reviewer #2
One issue they don't cover is the idea that a further point of signal integration mediated by GRKs might be reflected in receptor dimers and oligomers. How do things like barcodes in phosphorylation patterns get integrated into responses from GPCR heteromers? No one has seriously examined this issue- at least they could say that.
The reviewer is right, this is potentially an important issue. Unfortunately, there is too little unambiguous evidence regarding the role of class A GPCR oligomerization in their GRK/arrestin-regulation. All structures of G protein, GRK, and arrestin complexes with receptors solved so far reveal 1:1 interactions of GPCRs with potential signal transducers. We added a short section 3.3 (p. 12) to that effect.
The authors could also differentiate between a role for G proteins in recruiting GRKs and beta-arrestin- i.e. a structural role in GPCR signalling complexes and the need for activation of the G protein per se as in their discussion of biased agonists and the D2DR.
The reviewer is right, this is an important and quite controversial issue. As we recently reviewed this topic in detail, we only somewhat expanded its discussion (p. 16).
Minor Issues
There is some debate regarding whether or not PKA causes the beta2-AR to switch from Gs to Gi. The authors should nuance this to reflect the debate.
Thank you for drawing our attention to this issue! We added additional discussion and references (p. 3) to present a more balanced view.
Xenopus is misspelled on occasion.
Thanks! Corrected.